# Landslide Susceptibility Evaluation of Machine Learning Based on Information Volume and Frequency Ratio: A Case Study of Weixin County, China

**DOI:** 10.3390/s23052549

**Published:** 2023-02-24

**Authors:** Wancai He, Guoping Chen, Junsan Zhao, Yilin Lin, Bingui Qin, Wanlu Yao, Qing Cao

**Affiliations:** 1Faculty of Land Resources Engineering, Kunming University of Science and Technology, Kunming 650093, China; 2Key Laboratory of Geospatial Information Integration Innovation for Smart Mines, Kunming 650093, China; 3Spatial Information Integration Technology of Natural Resources in Universities of Yunnan Province, Kunming 650211, China

**Keywords:** landslide susceptibility, information value (IV), frequency ratio (FR), machine learning model, Weixin County

## Abstract

A landslide is one of the most destructive natural disasters in the world. The accurate modeling and prediction of landslide hazards have been used as some of the vital tools for landslide disaster prevention and control. The purpose of this study was to explore the application of coupling models in landslide susceptibility evaluation. This paper used Weixin County as the research object. First, according to the landslide catalog database constructed, there were 345 landslides in the study area. Twelve environmental factors were selected, including terrain (elevation, slope, slope direction, plane curvature, and profile curvature), geological structure (stratigraphic lithology and distance from fault zone), meteorological hydrology (average annual rainfall and distance to rivers), and land cover (NDVI, land use, and distance to roads). Then, a single model (logistic regression, support vector machine, and random forest) and a coupled model (IV–LR, IV–SVM, IV–RF, FR–LR, FR–SVM, and FR–RF) based on information volume and frequency ratio were constructed, and the accuracy and reliability of the models were compared and analyzed. Finally, the influence of environmental factors on landslide susceptibility under the optimal model was discussed. The results showed that the prediction accuracy of the nine models ranged from 75.2% (LR model) to 94.9% (FR–RF model), and the coupling accuracy was generally higher than that of the single model. Therefore, the coupling model could improve the prediction accuracy of the model to a certain extent. The FR–RF coupling model had the highest accuracy. Under the optimal model FR–RF, distance from the road, NDVI, and land use were the three most important environmental factors, ac-counting for 20.15%, 13.37%, and 9.69%, respectively. Therefore, it was necessary for Weixin County to strengthen the monitoring of mountains near roads and areas with sparse vegetation to prevent landslides caused by human activities and rainfall.

## 1. Introduction

Landslides refer to the movement of considerable rock, soil, or rock debris material along a slope, and they have been confirmed as one of the most devastating natural disasters worldwide [1,2]. In 2020, there were 4810 landslide disasters, 1797 rockslide disasters, 899 mudslide disasters, and 183 ground collapse disasters in China, thus resulting in 197 casualties. To be specific, landslide hazards account for over 60% of all geological hazards. The landslides and the secondary disasters have caused huge property damage and numerous casualties. This is especially true in mountainous areas that are characterized by complex geological environments, extreme weather, and the effect of human activities [3]. Many factors have been confirmed to contribute to the occurrence of landslides. Thus, analyzing the environmental factors for landslide occurrence, building a regional landslide susceptibility evaluation model, and evaluating the level of landslide susceptibility for landslide disaster prediction, prevention, and control, as well as land planning, are of great significance [4].

Landslide susceptibility evaluation methods have been primarily based on known landslide hazard data and GIS technology to construct landslide spatial prediction models for the quantitative analysis of potential landslides [5]. Scholars have evaluated landslide susceptibility in a variety of ways, and there are two main evaluation methods. The first type is based on knowledge-driven qualitative analysis [6]. The qualitative analysis follows an in-depth analysis of regional landslide causation mechanisms, using expert theory, knowledge, and experience to select landslide factors in the region and determine their weights for analysis [7,8]. The results of the evaluation are closely correlated with the evaluator’s experiential knowledge, which primarily comprises the fuzzy comprehensive evaluation method, the hierarchical analysis method [9], the expert scoring method [10], and the fuzzy comprehensive judging method [11,12,13]. The second method is mainly based on a data-driven quantitative analysis method. The above methods comprise mathematical and statistical models and machine learning models [14]. Mathematical statistical models refer to probability statistics or regression models for known landslide points. Subsequently, the entire study area is analyzed in accordance with the relative weights of the factors [15,16,17]. The mathematical statistical model is a probability statistic or a regression model for the known landslide points. Next, the entire study area is studied based on the relative weights of the factors. To be specific, information value model (IV) [18], weight of evidence model (WOE) [19,20], entropy index (IOE) [21], coefficient of determination (CF) [22,23,24], and frequency ratio model (FR) [25] are primarily involved in this model. With the increase in the volume of data, the complexity of topographic, geological, and hydrological elements cannot be fully resolved using simple mathematical and statistical methods of simulation and analysis. The machine learning model is to build a classifier that expresses all the data based on the properties of the existing training data, and the classifier optimally predicts all the data. In general, this model comprises logistic regression models (LR) [26], back propagation neural network models (BPNN) [27], support vector machine models (SVM) [28], random forests (RF) [29], Bayesian network models (BN) [30], and decision tree (DT) models [31]. Although some of the models have been employed for landslide susceptibility mapping in specific areas, no model has been proposed that can be applied to all landslide conditions. Scholars have begun to study the application of hybrid models in landslide susceptibility evaluation over the past few years to increase the accuracy of landslide prediction [22,32,33]. The landslide susceptibility evaluation method has been transformed from a single model to a hybrid model. Hybrid models combine two or more models to integrate landslide sample selection, feature selection, and information extraction into landslide hazard prediction [18,34,35,36]. To be specific, the merits of the different models can be dependent on each other to optimize the evaluation results and increase the prediction accuracy and can be applied to different geological conditions.

In the northeastern region of Yunnan Province, geological disasters occur frequently (e.g., landslides and debris flows). Accordingly, the landslide susceptibility evaluation in this region should be conducted for landslide hazard warning, prevention, and mitigation in this region. Most of the existing research on the evaluation of landslide susceptibility in the northeastern region of Yunnan Province has used a single evaluation model [37,38], rare research has coupled statistical models with machine learning methods, and there have been fewer opinions regarding landslide disaster prevention and control in the northeastern region of Yunnan Province. In this study, given topography, geological structure, hydrological environment, and land cover, 12 environmental factors are selected based on historical landslide hazards in Weixin County, with Weixin County in the northeastern region of Yunnan Province as an example. In this study, two statistical models (including an information volume model (IV) and a frequency ratio model (FR)) coupling three machine learning models (including logistic regression (LR), support vector machine (SVM), and random forest (RF)) are selected. The effectiveness and applicability of the coupling models in the region are studied, and the prediction accuracy of different coupled models is compared with that of a single model. A vulnerability zoning map with high accuracy is generated, thus providing a reference for disaster management and planning in northeastern Yunnan Province.

## 2. Materials and Methods

### 2.1. Study Area

Weixin County lying in northeastern Yunnan Province is part of Zhaotong City. It is located at the junction of Yunnan, Guizhou, and Sichuan provinces, with longitude and latitude between 104°41′15″ E to 105°18′45″ E and 27°42′30″ N to 28°07′30″ N, taking up a land area of 1400 km^2^. As depicted in Figure 1, the topographic elevation is low in the north and high in the south, with an elevation range of 480–1905 m, and the mountainous area accounts for 60% of the total area of the county. The county is densely populated with large and small rivers, and the Nanguang River, Baishui River, and Chishui River are the main water systems. The study area has a typical subtropical monsoon climate, with an average annual precipitation of 1076–1102 mm and an average annual temperature of 13.3 °C. The rainfall in Weixin County is mostly accumulated from May to October, accounting for 60–80% of the total annual rainfall. Geological and tectonic movements are strong, mainly in the Zangjiang Great Rift Zone, the Bijie Great Rift Zone, and the Chaomatian Rift Zone. The exposure strata in the study area primarily comprise the Middle–Upper Cambrian, Lower Middle Jurassic, Middle–Upper Permian, Terrace-phase Lower Middle Triassic, and Ordovician–Silurian strata; their lithology largely includes tuffs, mudstones, and sandstones with mud and gravel. The county’s land cover type is mainly forest land and arable land, accounting for 64% and 29.5% of the county’s total area, respectively. Landslide disasters occur frequently in the county due to the special topography of Weixin County and the complex and variable climatic conditions and abundant local rainfall.

Weixin County is one of the geological disaster-prone areas in Yunnan Province, which is one of the key areas for geological disaster prevention and control. In 2009, a major landslide occurs in Weixin County, resulting in 14 deaths and 12 missing persons, causing great damage to the lives and properties of residents. Landslide remote sensing images and field survey photos are shown in Figure 2.

### 2.2. Data Sources

#### 2.2.1. Landslide Cataloging Data

This study used GIS for data collection and processing. Landslide inventories are an important prerequisite for landslide susceptibility analysis because there is an assumption that past events have a strong influence on the future [39]. Thus, landslide inventory maps can provide useful information about the locations of previous landslides and may also identify areas where future landslides are likely to occur. In this study, a total of 345 landslide location point data were collected in Weixin County through the collection of historical data, remote sensing images, and Google image interpretation and field survey to construct a landslide inventory database. Then the landslide data were divided into a training dataset and a validation dataset for model building and validation, respectively. According to existing research [28,33,40,41], the dataset is usually divided according to 70% for training and 30% for validation [42,43]. The study area is mostly dominated by earthen landslides, and most of the landslides are traction landslides, a few are pile-fall landslides, and most of them are tongue-shaped and semicircular. The thickness of the landslides ranges from 1 m to 10 m, with small and medium-sized landslides predominating.

#### 2.2.2. Data Description of Environmental Factors

A landslide is one of the highly destructive natural hazards on the Earth’s surface [44]. The selection of the causal factors for landslides is a vital task for landslide susceptibility modeling and mapping. The environmental factors that induce landslides mainly comprise topography, geology, hydrology, and human engineering activities. The effective selection of environmental factors lays the basis for establishing landslide hazard susceptibility modeling and significantly affects the reliability and accuracy of evaluation results. According to the historical landslide disaster occurrence in Weixin County and field survey data, topography, geological structure, hydrology, and land cover were selected for the selection of environmental factors. The selected environmental factors are listed in Table 1. The raster cell data with a resolution of 30 m × 30 m were converted using ArcGIS 10.2 software. The data sources are listed in Table 1, and the environmental factors are presented in Figure 3.

## 3. Research Methodology

### 3.1. Research Technology Routes

The idea of this study was to couple two statistical methods (IV and FR) with three machine learning methods (LR, SVM, and RF) to build nine landslide susceptibility evaluation models, to conduct a comparative analysis of the performance of single and coupled models, and finally to analyze the contribution of environmental factors to landslide development under the optimal model. The procedures are presented in a flowchart as shown in Figure 4 and included the following steps: 

Step 1 was to collect landslide hazard-related data in the study area, mainly including historical landslide hazard site data and environmental factors data.

Step 2 was to perform independence tests for environmental factors, mainly Pearson correlation coefficients and multicollinearity diagnostics.

Step 3 was to obtain the frequency ratio and information value of the respective environmental factor by the frequency ratio and information value methods, respectively, and then obtain the influence law of each environmental factor on landslide development in its attribute interval.

Step 4 was to divide historical landslide hazard points, randomly select landslide and non-landslide points according to the ratio of 7:3, obtain training and validation sets, build nine landslide susceptibility prediction models (LR, SVM, RF, IV–LR, IV–SVM, IV–RF, RF–LR, FR–SVM, and FR–RF), and build landslide susceptibility maps basing on GIS.

Step 5 was to analyze and compare the performance of the models based on the confusion matrix, ROC curves, and AUC values of the validation dataset to find the optimal model.

Step 6 was to discuss the significance of the respective environmental factor based on the optimal model and rank the contribution of the respective factor to obtain the important trigger factors for landslide susceptibility in the study area.

### 3.2. Screening of Environmental Factors

Landslides are affected by a variety of factors (e.g., intrinsic and extrinsic factors), and there is a certain correlation between the factors. The extremely high correlation between factors will lead to problems (e.g., complexity of model operation and model overfitting). Accordingly, the correlation analysis between the evaluation factors should be conducted before establishing the evaluation model to eliminate the factors with high correlation to ensure the efficiency of the model operation and the rationality of the evaluation results [32,45]. Thus, the Pearson correlation coefficient (PCC), variance inflation factor (VIF), and tolerance level (TOL) were employed for independence tests.

#### 3.2.1. Correlation Analysis of Factors

The Pearson correlation coefficient (PCC) method is a nonparametric statistical method that is adopted to measure the correlation (linear correlation) between two variables *X* and *Y*, with a value between −1 and 1. In general, the correlation coefficient is expressed as r. When r > 0, the two variables are positively correlated; when r < 0, the quantitative variables are negatively correlated; and when r = 0, the two variables are not correlated. In general, the Pearson coefficient method only refers to the absolute value of the correlation coefficient r to indicate the correlation level between two variables. The absolute value r ranging from 0 to 0.3 indicates uncorrelation, |r| between 0.3 and 0.6 indicates low correlation, |r| between 0.6 and 0.8 indicates moderate correlation, and |r| between 0.8 and 1.0 indicates high correlation [46]. The calculation equation is as follows:(1)$$r=\frac{\sum_{i=1}^{n}\left(X_{i}-\bar{X})(Y_{i}-\bar{Y}\right)}{\sqrt{\sum_{i}^{n}{\left(X_{i}-\bar{X}\right)}^{2}}\sqrt{\sum_{i}^{n}{\left(Y_{i}-\bar{Y}\right)}^{2}}}$$
where r denotes the correlation coefficient, $X_{i}$ and $Y_{i}$ are the observation values of point i corresponding to variables X and Y, and $\bar{X}$ $\bar{Y}$ denote the sample means of X and Y, respectively.

#### 3.2.2. Multicollinearity Tests

Multicollinearity refers to the existence of a certain linear relationship between explanatory variables in a multiple regression model. Variance inflation factor (VIF) and tolerance level (TOL) are commonly used to test for multicollinearity. VIF takes a value higher than 1, and TOL takes a value between 0 and 1. When the VIF value is higher than 2 or TOL is less than 0.5, it indicates that there is strong multicollinearity between the factors; when VIF is higher than 1 and less than 2 or TOL is higher than 0.5, the multicollinearity between the factors is light [47]. The calculation is as follows:(2)$$\mathrm{VIF}=\frac{1}{1-R_{i}^{2}}=\frac{1}{\mathrm{TOL}}(i=1,2,3\cdot \cdot \cdot \cdot \cdot \cdot k)$$
where $R_{i}$ denotes the correlation coefficient when the independent variable $X_{i}$ is a regression coefficient on the remaining variabsles.

### 3.3. Processing of Factors

#### 3.3.1. Frequency Ratio (FR)

Frequency ratio (FR) method is a binary statistical method adopted to explain the probabilistic relationship between the dependent and independent variables [8,16,48]. The frequency ratio is capable of calculating the correlation between the probability of landslide occurrence and the evaluation factors. The effect of different levels of the respective factor on the occurrence of landslide (e.g., the contribution rate) is analyzed in accordance with the value of frequency ratio [16,20]. In this study, the frequency ratio method was employed to quantitatively analyze the correlation between the analysis of landslide distribution and the evaluation factors. The size of the FR value can reveal the problem of the contribution rate of the respective attribute interval of environmental factors to landslide generation, the larger the FR value [20]. The above analysis suggested that it more significantly contributed to the occurrence of landslide, and vice versa, that it was difficult for the landslide to occur in the interval. The frequency ratio equation is as follows:(3)$$\mathrm{FR}=\frac{N_{i}/N}{S_{j}/S}$$
where $N_{j}$ denotes the number of raster cells of the evaluation factor with landslides in interval i, N represents the number of raster cells of landslides in the whole area, $S_{j}$ expresses the number of raster cells of the evaluation factor in this graded interval, and $S_{j}$ denotes the number of raster cells in the whole area.

#### 3.3.2. Information Volume (IV)

The information volume model (IV) is a common statistical method originally used in the field of mineral resource census exploration [14,17], etc. The magnitude of information is adopted to quantitatively describe the contribution of the respective factor to regional mineralization. Subsequently, some scholars have used the informativeness model for geological hazards [21]. In the evaluation of landslide susceptibility, the information size characterizing the occurrence of landslides under the attribute interval of the respective influence factor is obtained using the density of landslide occurrence based on the statistical historical landslide data [25]. The contribution rate of each factor attribute interval to landslide occurrence is judged based on the information size. The information size is calculated as follows:(4)$$\mathrm{IV}(x_{i},H)=\ln\left(\frac{A_{i}/A}{B_{i}/B}\right)$$
where $I(x_{i},H)$ denotes the value of information provided by the occurrence of landslide hazard, $x_{i}$ represents the rank of indicator factor in the evaluation unit, $A_{i}$ expresses the number of landslide raster cells of indicator factor in the study area, A is the total number of landslide raster cells in the study area, and B is the total number of raster cells of indicator factor in the study area. IV < 0, thus suggesting that the landslide was unfavorable under the level of evaluation factor i. IV > 0, thus revealing that the evaluation factor was unfavorable to the occurrence of landslide under the interval of attribute i.

### 3.4. Machine Learning Models

#### 3.4.1. Logistic Regression

Logistic regression (LR) is a regression analysis method that has been usually adopted to explain the correlation between dichotomous dependent variables or predictor variables [3]. The logistic regression is compared with the general linear regression model in that the above variables can be continuous or discrete variables [10]. Thus, LR is a dichotomous problem by predicting the probability of an event occurring (“0” and “1”). The LR is expressed as follows:(5)$$P=\frac{e^{Y}}{1+e^{Y}}$$
(6)$$Y=\alpha +\beta_{1}X_{1}+\beta_{2}X_{2}+\cdot \cdot \cdot +\beta_{n}X_{n}$$
where Y represents the denotes landslide event occurrence, α is a constant, *P* expresses the probability of a landslide, and $X_{i}$ represents the influence factors for the occurrence of landslides.

#### 3.4.2. Support Vector Machines

Support vector machines (SVM) are a supervised learning method developed in accordance with statistical theory and the principle of structural risk minimization [22,49]; these machines have been extensively applied to classification and regression. The support vector machine is a binary classification model based on the principle of finding the separating hyperplane that maximizes the classification interval in a high-dimensional space [31,34,35]. The input data are $x_{i}(i=1,2,3,\cdot \cdot \cdot n)$, and the output of the corresponding binary classification problem is (y=±1). The calculation is written as Equations (7)–(9):(7)$$L=||\omega |{|}^{2}/2-\sum_{i}^{N}\lambda_{i}((y_{i}(\omega \cdot x_{i})+b)-1)$$
(8)$$y_{i}=(\omega_{i}\cdot x_{i})+b\geq 1$$
(9)$$y_{i}(\omega \cdot x_{i})+b\geq 1+\zeta_{i}$$
where $||\omega |{|}^{2}$ denotes the norm of the normal vector in the hyperplane, b is a constant, L expresses the loss function, $\lambda_{i}$ is the Lagrange multiplier, and $\zeta_{i}$ is the relaxation factor. In Equation (5), v(0,1] denotes the misclassification, and the radial kernel function is selected as the kernel function of SVM in this study. The formula is as follows:(10)$$L=\frac{1}{2}||\omega |{|}^{2}-\frac{1}{vn}\sum_{i=1}^{n}\zeta_{i}$$

#### 3.4.3. Random Forests

Random forest (RF) is a machine learning algorithm proposed by Leo Breiman, belonging to the type of integrated Bagging algorithm [29]. Many decision classification trees are randomly generated, and the respective decision tree can be voted or averaged to select the optimal classification result, allowing the model to analyze the results with high accuracy and generalization [50]. The principle of random forest is to generate a novel decision tree of training samples by randomly electing n samples from the original training dataset N by the self-service (bootstrap) resampling technique and then to repeat the above steps to generate m decision trees to form a random forest [8,28]. Moreover, the classification results of the novel data are determined by the number of votes formed by the classification trees. Random forest is a modification of the decision tree algorithm by combining multiple decision trees together, with the creation of the respective tree depending on the independently drawn samples [30,41]. The classification power of a single tree may be small, whereas a test sample can statistically select the most likely classification by the classification results of each tree after considerable decision trees are generated randomly. Random forest exhibits strong noise immunity stable performance, can handle high dimensionality, and does not have to do feature selection.

### 3.5. Performance Validation of the Model

#### 3.5.1. Confusion Matrix

In this study, the performance of the evaluation model for landslide susceptibility was evaluating using the confusion matrix, which is often used in binary classification for the evaluation of model performance. The confusion matrix included four common parameters as listed in Table 2. In this study, four statistical indicators (including precision, recall, accuracy, and F1 score) were used to evaluate the performance of the respective model. The indicators are expressed as follows:(11)$$\mathrm{Precision}=\frac{\mathrm{TP}}{\mathrm{TP}+\mathrm{FP}}$$
(12)$$\mathrm{Recall}=\frac{\mathrm{TP}}{\mathrm{TP}+\mathrm{FN}}$$
(13)$$\mathrm{Accuracy}=\frac{\mathrm{TP}+\mathrm{TN}}{\mathrm{TP}+\mathrm{FP}+\mathrm{FN}+\mathrm{TN}}$$
(14)$$F1\text{-score}=\frac{2\times \mathrm{Precision}\times \mathrm{Recall}}{\mathrm{Precision}+\mathrm{Recall}}$$
where TP denotes the number of landslide rasters predicted correctly, TN denotes the number of non-landslide rasters predicted correctly, FP denotes the number of landslide rasters predicted incorrectly, and FN denotes the number of non-landslide raster cells predicted incorrectly.

#### 3.5.2. ROC Curves and AUC Values

Recipient characteristic curves are often used in assessing landslide susceptibility evaluations. Receiver operating characteristic (ROC) curves are also a measure of model validity [51]. The area of the graph enclosed by the curve and the axes is called the area under the curve AUC. The AUC values range from 0.5 (worst model performance) to 1 (optimal model performance) [14,52]. When the AUC value is higher than 0.7, the closer the AUC value is to 1, the more accurate the model’s prediction is. The value of AUC can be calculated by the integral trapezoidal rule. The equation is written as follows:(15)$$\mathrm{AUC}=\frac{\left(\sum \mathrm{TP}+\sum \mathrm{TN}\right)}{\left(P+N\right)}$$
where TP (true) and TN (true negative) denote the correctly classified raster cells, P expresses the total number of landslide raster cells, and N represents the total number of non-landslide raster cells.

## 4. Results

### 4.1. Independence Test of the Factors

#### 4.1.1. Pearson’s Correlation Tests

Existing research has suggested that the environmental factors selected for model construction should maintain relative independence from each other to ensure the accuracy of model evaluation [32]. The correlation test of the factors was performed using the statistical tool of band set in ArcGIS, and the correlation coefficient matrix of 14 environmental factors was yielded and visualized using Origin software. The results are illustrated in Figure 5. When the absolute value of the correlation coefficient between two factors r > 0.6, the factors are considered to be strongly correlated with each other. As depicted in Figure 5, the correlation coefficients of the factors were all less than 0.6, and the correlations were weak. Accordingly, the factors had a small degree of interaction.

#### 4.1.2. Multicollinearity Tests

In addition, multiple covariance analysis was conducted on the environmental factors using SPSS 22 software to obtain the TOL and VIF values of the environmental factors. The results are listed in Table 3; all environmental factors achieved TOL values less than 1 and VIF values less than 2. The above result indicated that there was no covariance among the selected environmental factors, thus verifying the rationality among the selected environmental factors.

### 4.2. Classification of the Respective Attribute Interval of Environmental Factors and Calculation of Frequency Ratio and Information Value

In this study, the ratio of the total number of landslide rasters to the total number of rasters was used for calculation.

(1) Factors for topography and geomorphology were as follows: The distribution of landslides is closely correlated with the elevation, and the vegetation cover and soil moisture are different in different elevation areas, which leads to different surface water collection capacity. Moreover, the intensity of human activities varies in different elevation ranges [53]. As depicted in Table 4, the information values between elevations 478–800 m, 1000–1100 m, and 1200–1300 m were higher than 0, and the frequency ratio was higher than 1, indicating that landslides were densely distributed in this interval. The slope affected the development and gestation process of the landslide by affecting the stress distribution, surface runoff, and groundwater recharge and discharge of the slope body [54]. Slope orientations between 20°–25°, 25°–30°, and 30°–40°, with the information values higher than 0 and frequency ratios higher than 1, accounted for more than 70% of the landslides occurring. The slope orientation was influenced by the intensity of solar radiation, leading to differences in evaporation, vegetation, and human activities [55]. In the east, south, west, and southwest directions, landslides were prone to occur, with frequency ratios higher than 1 in all the above directions and positive information values. The degree of twisting and deformation of slope surfaces affected the stress distribution in the slope parts and thus had different degrees of influence on the development of landslides [56]. In this study, plane curvature and profile curvature were chosen. As depicted in Table 5, the profile curvature ranged from 0.35 to 4.92 degrees, and the profile curvature ranged from −12.33 to −0.15 and 2.47 to 15.42. The frequency ratios were all higher than 1, and the information values were all higher than 0, indicating that this interval was favorable for landslides to occur.

(2) Factors for hydrological environment were as follows: Rainfall is one of the main triggering factors for landslide development, and rainfall changes the stress distribution of slope body and the stability of slope body [57]. The rainfall was between 1084 and 1095 mm, the information value was higher than 0, and the frequency ratio was higher than 1. Landslides occurred more frequently in this zone. Within a certain distance from the water system, the scouring and soaking of rivers could result in the loss of soil from the slope, thus leading to the destabilization of landslides. As depicted in Table 4, the FR value was higher than 1, and the IV value was higher than 0 within the distance of 800 m from the water system, indicating that landslides were likely to occur in this area.

(3) Factors for geology were as follows: In the range of distance less than 1200 m from the fracture zone, the FR value was higher than 1, and the information value was higher than 0, indicating that the distance from the fracture zone affected the occurrence of landslides. The stratigraphic lithology was an important internal factor for landslide development and stability [58]. The physical and mechanical properties of the rock mass and the interstratigraphic mechanism determined the geotechnical stress distribution, which in turn affected the stability of the slope. In this study area, landslides occurred mainly in the Middle and Upper Permian sandstones and carbonaceous rocks, etc. Landslides easily occurred in the lithological distribution area, as depicted in Table 4.

(4) Factors for land cover were as follows: The ground cover of vegetation affected the development and distribution of landslides, which mainly played a certain fixed role on the slope surface through the rhizomes of vegetation and slowed down the water flow rate and infiltration rate of the slope surface. Landslides easily occurred in areas with low vegetation cover, and their FR values were higher than 1 and IV values were higher than 0. Irrational human exploitation of land is also one of the causes of landslides, and different land cover types have different effects on the stability of slopes [59]. As depicted in Table 5, landslides occurred more in the range of construction land, water area, and grassland. Their FR values reached up to 4.65, 2.62, and 1.68, respectively, and the information values were positive. The distance from the road was also a vital factor for the development and distribution of landslides. Roadbed widening, blasting works, and artificial slope cutting in road projects could lead to the changed original geomorphology and geotechnical structure, thus resulting in the reduced slope stability. As depicted in Table 4, at the distance from the road less than 400 m, the FR values were obtained as 3.01 and 1.01, respectively, and the information values were higher than 0, thus revealing that the closer the distance from the road, the greater the possibility of landslide occurrence will be.

### 4.3. Results of the Model

#### 4.3.1. LR Regression and Coupling Model

In this study, the original and frequency ratios and information values of the respective environmental factor in the training sample were input into SPSS 22 software for binary logistic regression calculation, and the regression coefficients and constants of each environmental factor were obtained. The magnitude of the regression coefficients could indicate the degree of contribution of the evaluation index factors to landslide generation and the regression coefficients under different models, as listed in Table 5. Next, the regression coefficients were substituted into Equations (5) and (6) and calculated using ArcGIS 10.2 raster calculator to predict the landslide sensitivity index of the respective raster cell in the study area.

#### 4.3.2. Support Vector Machine (SVM) and Coupling Model

In this study, the data of the training set and the test set were input into Python, built using the Scikit-learn framework, then input into the svm library. Moreover, the regularization parameter ϑ and gamma parameter values of the support vector machine model RBF kernel function were obtained based on the training dataset using tenfold cross-validation and the optimal grid. The single SVM penalty parameter reached 0.4, the gamma parameter was 0.114, and the regularization parameters of the IV–SVM and FR–SVM models based on information value (IV) and frequency ratio (FR) coupling were 0.75, 0.24 and 0.83, 0.12, respectively. Subsequently, the trained models were employed for landslide sensitivity prediction for the whole study of 1,548,205 point objects. Afterward, the sensitivity values of all points were input into ArcGIS and then converted into 30 × 30 raster cells.

#### 4.3.3. Random Forest and Coupling Models

In this study, the training set and test set data were input into Python built using the Scikit-learn framework and input into the RandomForestClassifier library. n_estimators and max_depth, important parameters in random forest, significantly affected the accuracy of the model. n_estimator represents the number of decision trees. The prediction performance of the RF model improved with the increase in n_estimators, whereas the computational effort of the model tended to increase, and the modeling time was extended. The parameters of the RF model alone were 150, and the parameters of the coupled models IV–RF and FR–RF based on information value (IV) and frequency ratio (FR) were 100 and 104, respectively. The trained model was then used for landslide sensitivity prediction of 1,548,205-point objects in the whole study area, and the sensitivity values of all points were input into ArcGIS10.2 and then converted into 30 × 30 raster cells.

### 4.4. Landslide Susceptibility Mapping

In accordance with Section 4.3, the landslide susceptibility map of Weixin County was generated (Figure 6) using the frequency ratio (FR), information value (IV), coupling logistic regression (LR), support vector machine (SVM), and random forest (RF) models. The landslide susceptibility zoning was based on the probability of landslide occurrence in the study area obtained from the models. The susceptibility of the study area was divided into five main classes: very low, low, medium, high, and very high susceptibility zones. The mapping of the landslide susceptibility results obtained from the nine models was relatively similar, which was generally consistent with the results of the field survey.

(1) Very high and high susceptibility areas often caused a large number of landslides due to vegetation destruction, unreasonable land use, road construction, housing development, and other reasons. Therefore, the very high landslide susceptibility areas in Weixin County were mainly distributed linearly along rivers and road extension areas and in areas with more frequent human activities. As shown in Figure 4, the very high and high susceptibility areas in Weixin County were mainly located in the western, southeastern, central, and northwestern parts of the study area. The low and very low susceptibility areas were mainly located in the northern part of the study area where there was high forest cover and less human activity, which was consistent with the results of the field survey.

(2) In the western, central, and southeastern parts of the study area, the very high susceptibility zones were mainly in the Ordovician, Middle Cambrian, and Cenozoic Quaternary periods, with the main lithologies being dolomite, shale, mudstone, and fine sandstone, etc. Under the influence of weathering, the rock bodies in the weathered crust layers were relatively fragmented, especially at the lithological boundaries of the strata. The loose structure of the soil layers during this period provided abundant material for landslides to occur. The unstable stratigraphic structure in the stratigraphic lithological boundary area was an important factor influencing the occurrence of landslides.

(3) In granitic rock areas, very high susceptibility areas were more clearly distributed mainly along roads. Environmental factors such as humidity, topographic relief, and geological tectonic activity accelerated weathering, altering the inherent nature of the material and reducing the strength of the surface rocks.

(4) The historical landslide data were overlaid with the landslide susceptibility zoning results for analysis, and the area shares, landslide shares, and frequency ratios of both for the respective zoning were calculated. The statistical results are listed in Table 6. With the increase in the landslide susceptibility level, the landslide percentage and the frequency ratio percentage increased, and the landslides in significantly high and high susceptibility areas accounted for over 70% of the total number of historical landslides. The areas of the significantly high susceptibility zone based on single models (LR, SVM, and RF) were 134.17 km^2^, 124.98 km^2^, and 113 km^2^, respectively; their proportions of the study area reached 9.63%, 8.97%, and 8.12%, respectively; and the frequency ratios were 4.01, 6.31, and 7.05, respectively. Based on coupled models of FR, the significantly high susceptibility areas achieved IVs (IV–LR, IV–SVM, IV–RF, FR–LR, FR–SVM, and FR–RF) of 120.45 km^2^, 121.71 km^2^, 115.58 km^2^, 120.42 km^2^, 122.60 km^2^, and 114.17 km^2^, respectively. Moreover, their proportions of the study area were obtained as 8.64%, 8.74%, 8.29%, 8.64%, 8.80%, and 8.19%, and the frequency ratios were 4.73, 4.65, 6.60, 6.49, 6.92, and 7.21, respectively. In brief, the area ratios of the significantly high, high, medium, low, and very low susceptibility zones in the study area were consistent with the distribution pattern of landslide hazards.

The three single machine learning models and the landslide susceptibility maps obtained by coupling FR and IV models had similar changing trends. Figure 6 shows that the significantly high and high susceptibility zones were primarily distributed in the distance from rivers, road extension areas, and areas with more frequent human activities, with the low vegetation cover, the well-developed water system, as well as the soft rock lithology distributed along the fracture zone. In the southeastern region, the significantly high susceptibility zone obtained by the support vector machine and its coupled model was larger in scope. As revealed by the actual survey data and images, the soil was loose and the ecological environment was fragile in this area, and there was a great possibility of geological disasters (e.g., landslides).

### 4.5. Accuracy Evaluation of the Model

The frequency ratio of the respective model was calculated by counting the distribution of historical landslide raster cells in each susceptibility class. The results are listed in Table 7. The results showed that 83% of the landslide raster cells fell into the significantly high and high susceptibility zones in the coupled FR–RF-based model, and more than 80% of the raster cells fell into the significantly high and high susceptibility zones in the remaining coupled models. The percentages of landslide raster cells falling into the significantly high and high susceptibility zones in the single model were the lowest in the LR model, the second highest in the SVM model, and the highest in the RF model. The above analysis shows that the prediction accuracy of the coupled model was higher than that of the single model in general.

#### 4.5.1. Evaluation of Precision Parameters

The results of the confusion matrix and statistical indicators are listed in Table 7; each index in shown in Figure 7. In terms of precision, FR–RF > IV–RF > RF > SVM > IV–SVM > FR–SVM > IV–LR > FR–LR > LR; the FR–RF model had the highest precision, indicating that the FR–RF model had the strongest partitioning ability for negative samples. In terms of recall, FR–RF > IV–RF > RF > FR–SVM > IV–SVM > SVM > FR–LR > IV–LR > LR. It can be seen that the FR–RF model had the highest recall, which indicated that the FR–RF model had the strongest ability to identify positive samples. Second, in terms of F1 scores, the F1 scores of all models were higher than 0.7, and the ranked values of the F1 scores of the respective model were FR–RF > IV–RF > RF > FR–SVM > SVM > IV–SVM > FR–LR > IV–LR > LR, which indicated that all models could reflect the landslide susceptibility of the study area, and the performance of the FR–RF model was relatively higher. In terms of accuracy, RF > SVM > LR, indicating that the RF model could predict the occurrence of landslide hazards better than the SVM and LR models, followed by the highest accuracy of the FR–RF model, indicating that the coupled model could improve the prediction accuracy of the model.

#### 4.5.2. Comparison of ROC Curve and AUC Values

Figure 8 presents the operating characteristic curves of the subjects for the nine models (including LR, SVM, RF, FR–SVM, FR–RF, FR–LR, IV–SVM, IV–RF, and IV–LR). The LR, SVM, and RF models achieved AUC values of 0.761, 0.855, and 0.936, respectively, and the AUC values achieved by the IV–LR, IV–SVM, and IV–RF models reached 0.791, 0.867, and 0.927, respectively. The AUC values achieved by the FR–LR, FR–SVM, and FR–RF models were 0.785, 0.885, and 0.949, respectively. The AUC values of all models are evaluated to be above 0.75. To be specific, the RF model exhibited the highest accuracy among the single models, the LR model achieved the lowest accuracy, and the FR–RF model had the highest accuracy among the coupled models, followed by the IV–RF model. Furthermore, the lowest AUC value achieved by the coupled FR–LR model was higher than that of the single model, thus revealing that the coupled model was beneficial to enhance the prediction ability of landslide hazard.

### 4.6. Case Study

The prediction performance of the model was analyzed and verified by using 15 landslide data points obtained from interpretation and investigation in 2021–2022. The FR–RF model with the best prediction effect was selected, and 10 landslides were found in the extremely high and highly prone areas of the model. From this point of view, the coupled model had good predictive performance. The Longdong Rock landslide in Guihua Village, Zhaxi Town, was selected, and the results showed that the landslide fell well into the extremely prone area of FR–RF model, as shown in Figure 9. It was found in the field investigation that the landslide was developed in the weathering mud dolomite rock mass in Paleozoic Ordovician (O1–S1). Due to the combined action of lithology, stress change in the slope after artificial excavation of slope foot, and groundwater, the rock mass weathering and slime phenomenon in this layer were obvious, and the physical and mechanical properties of the soil mass were low. Finally, the landslide developed into a slip zone and caused the slope to slide.

## 5. Discussions

The landslide susceptibility of Weixin County was evaluated using remote sensing, GIS tools, and machine learning algorithms. Three single models (including LR, SVM, and RF) and the hybrid models (IV–LR, IV–SVM, IV–RF, FR–LR, FR–SVM, and FR–RF) were built based on the information value (IV) and frequency ratio (FR) to draw nine landslide susceptibility maps in Weixin County, and the prediction accuracy of the nine models was compared. The comparison results indicated that the prediction accuracy of the coupled IV and FR-based models was higher than the single model accuracy. The hybrid model with frequency ratio coupled with random forest achieved the highest prediction accuracy among all models. The coupled models based on the amount of information value and frequency ratio exhibited higher prediction accuracy than the single model. They took on a critical significance to predicting possible future landslides and laid a basis for decision-making in the early warning and prevention of landslides in Weixin County.

### 5.1. Landslide Susceptibility Map Rationality

Through superposition analysis, the rationality of landslide susceptibility mapping in Weixin County was evaluated. As shown in Table 7, the detailed information of different levels of prone areas in different models was presented. Landslide susceptibility maps based on single models and coupled models of information content and frequency ratio had the same trend on the whole. In the LR, IV–LR and FR–LR models, the proportions of extremely highly prone areas were 8.64%, 8.64%, and 9.63%, and the proportions of landslides were 38.5%, 40.91%, and 40.14%, respectively. The frequency ratios were 4.01, 4.73, and 4.65. In SVM, IV–SVM, and FR–FR models, the proportion columns of extremely highly prone areas were 8.97%, 8.74%, and 8.8%, and the proportions of landslides were 56.62%, 57.62%, and 57.11%, respectively. The frequency ratios were 6.31, 6.6, and 6.49. In the RF, IV–RF, and FR–RF models, the proportions of extremely highly prone areas were 8.12%, 8.29%, and 8.19%; the proportions of landslides were 57.11%, 57.66%, and 59.11%; and the proportions of frequency were 7.05, 6.92, and 7.21. The above analysis showed that the landslide susceptibility map of Weixin County obtained by most models was reasonable, and the proportion of landslide occurrence gradually increased from very slightly prone areas to very highly prone areas. In general, compared with other models, the landslide susceptibility map obtained by the RF–RF model was the most reasonable. The above model achieved good results in the landslide susceptibility assessment mapping of Weixin County. In this study, only statistical methods and typical cases were selected to analyze and verify the results of the result analysis. In future studies, the susceptibility mapping results will be analyzed from the perspective of geological concepts, so as to make the landslide susceptibility mapping results more accurate and reliable.

### 5.2. Evaluation Units

The accuracy of landslide susceptibility evaluation was closely related to the selection of the evaluation unit. The commonly used evaluation units mainly include the grid unit, slope unit, topographic and geomorphic unit, and administrative unit. After selecting the appropriate evaluation unit, each evaluation unit can assign the value of each environmental factor. The grid unit is a grid that divides the study area into regular grids for storage and calculation. This method is widely used in landslide susceptibility assessment mapping, but the grid unit could not fully reflect the topographic relief and geological and hydrological elements of the study area. A grid cell size of 30 m × 30 m was used in this study, but in future studies, slope cells can be considered for landslide susceptibility analysis, and the similarities and differences between slope cells and grid cells can be compared.

### 5.3. Significance of Environmental Factors

The selection of suitable environmental factors is of great importance to landslide susceptibility evaluation. However, the selected evaluation factors are not all strong predictors; in some cases, some factors will generate noise and lead to the decreased accuracy of prediction. In the existing research, the preliminary analysis of the correlation between the respective environmental factor and landslide is only conducted, and the correlation interval of the occurrence of the respective environmental factor is obtained, whereas the contribution of each environmental factor to landslide susceptibility is not revealed [14,60]. Section 4.3 indicates that FR–RF with the highest model accuracy was selected for the factor significance analysis. In this study, the significance of the environmental factors was measured as the percentage decrease in the average Gini index versus the sum of the decrease in the average Gini index for all environmental factors. The 12 environmental factors were analyzed using the Python language and then visualized with Origin 2021 software to generate the significance ranking graph of the respective factor, and the results are illustrated in Figure 10. As depicted in Figure 10, the 12 environmental factors were ranked in order of significance as follows: distance to roads > NDVI > land use > stratigraphic lithology > elevation > rainfall > slope direction > distance to rivers > distance from fracture zone > slope > plane curvature > profile curvature. Distance to roads, NDVI, and land use were the three critical environmental factors, and their significance percentages were 20.15%, 13.37%, and 9.69%, respectively. It indicated that the above three factors contributed the most to the model and were important triggers for landslide generation in the study area. The lowest significance percentages of planar curvature and profile curvature were 2.80% and 1.79%, respectively, indicating that the above two environmental factors had a weak influence on the evaluation of landslide susceptibility in the study area.

### 5.4. Uncertainty of the Coupling Models

Hybrid models of statistical methods and machine learning methods have been increasingly applied to landslide susceptibility evaluation, which significantly increases the prediction accuracy of the models. Statistical methods are vital links between landslide susceptibility indices and environmental factors, and their linkage performance takes on a critical significance to the prediction accuracy of machine learning models. The commonly used statistical methods are deterministic factor, weight of evidence, information value, entropy index, and frequency ratio. The current research has not specified which statistical methods can improve the prediction accuracy of machine learning models. Different statistical methods bring great uncertainty to the combination of machine learning methods for landslide susceptibility prediction. In this study, only information values and frequency ratios were selected using three machine learning methods (including logistic regression, support vector machine, and random forest). In future research, more statistical methods and machine learning methods will be employed to analyze the uncertainty patterns in landslide susceptibility prediction.

## 6. Conclusions

Landslide susceptibility mapping is a key link in landslide hazard control. This study used Weixin County of China as the research area and selected appropriate environmental factors according to the data of historical landslide disaster points as the basic data. Three single models (LR, SVM, and RF) and coupled models (IV–LR, IV–SVM, IV–RF, FR–LR, FR–SVM, and FR–RF) based on the information volume (IV) and frequency ratio (FR) were constructed to carry out landslide susceptibility evaluation in Weixin County and generate landslide susceptibility map. The accuracy of the model was evaluated by various statistical indexes, and the accuracy of the model was evaluated by the ROC curve. In summary, the main conclusions were as follows: (1) The landslide susceptibility map obtained by the single models LR, SVM, and RF and the coupling models based on IV and FR (IV–LR, IV–SVM, IV–RF, FR–LR, FR–SVM, and FR–RF) had a good effect. The areas with high landslide hazard and high landslide risk in Weixin County were mainly distributed in the west, southeast, central, and northwest regions, extending along roads and rivers. The very low and low susceptibility areas were mainly distributed in the northern mountainous areas with fewer human activities and the southern areas with higher forest coverage. The accuracy and reliability of the model were verified by statistical index parameters, and the accuracy of the coupled model was higher than that of the single model on the whole. (2) The ROC curve, AUC value, and statistical index were used to evaluate and compare model performances. The overall accuracy of susceptibility based on single models was lower than that of coupled models. The FR–RF coupling model had the highest accuracy, and the AUC value was 0.949. Under the optimal FR–RF model, the importance analysis of environmental factors needs to strengthen the monitoring of mountains near roads and areas with sparse vegetation to prevent the occurrence of landslides caused by human activities and natural rainfall.

This study described in detail the construction of a single machine learning model and a coupled model based on the information content (IV) and frequency ratio (FR) and compared the performance between the models. The accuracy of the coupled model was further verified in this paper. In addition, this study can provide the government decision-making efficiency of landslide prevention and control, which is conducive to the rapid response of landslide warning. Integrated risk assessors and land use planning could also benefit from our findings.

## Figures and Tables

**Figure 1 sensors-23-02549-f001:**
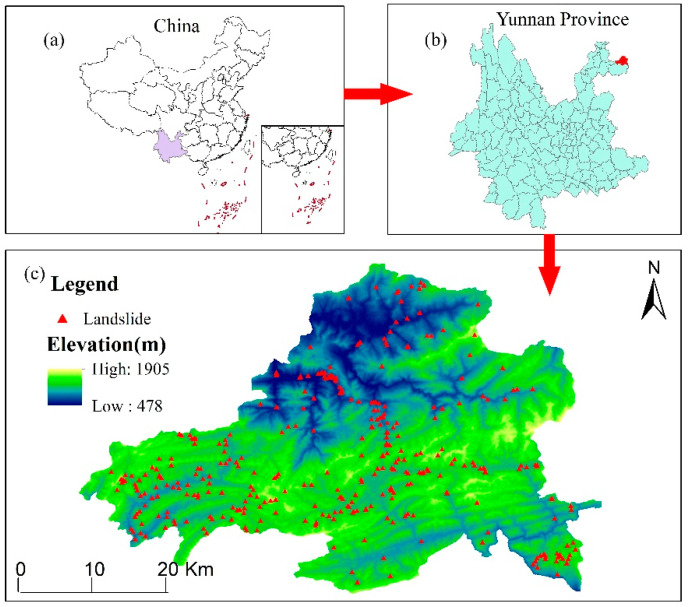
Location of the study area and landslide inventory. (**a**) The regions of China, small boxes indicate local areas. (**b**) In Yunnan Province, the red part is Weixin County. (**c**) Location of landslide in Weixin County.

**Figure 2 sensors-23-02549-f002:**
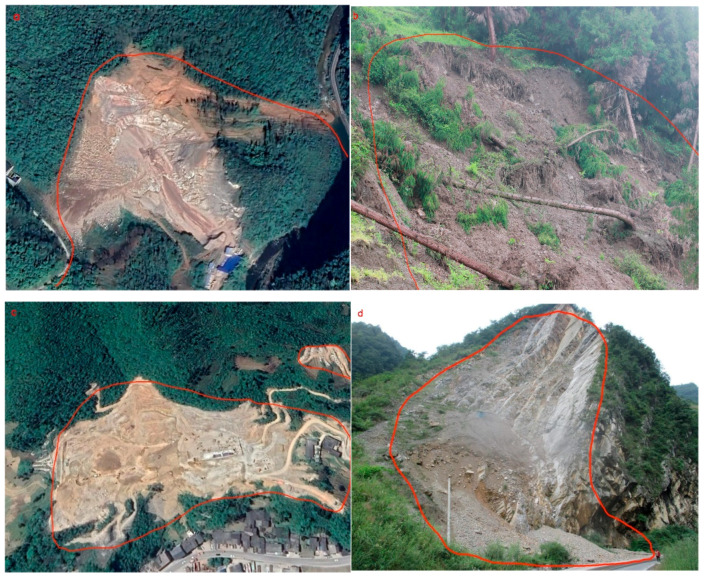
(**a**,**c**) Examples of landslides from Google Earth and their locations marked in Figure 1. (**b**,**d**) Examples of landslides from field investigation and their locations marked in Figure 1.

**Figure 3 sensors-23-02549-f003:**
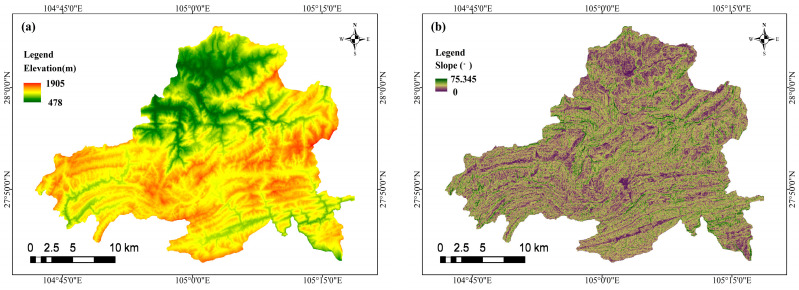
The environmental of factors: (**a**) elevation, (**b**) slope, (**c**) aspect, (**d**) distance to roads, (**e**) distance to rivers, (**f**) distance to faults, (**g**) rainfall, (**h**) land use, (**i**) lithology, (**j**) NDVI, (**k**) plan curvature, and (**l**) profile curvature.

**Figure 4 sensors-23-02549-f004:**
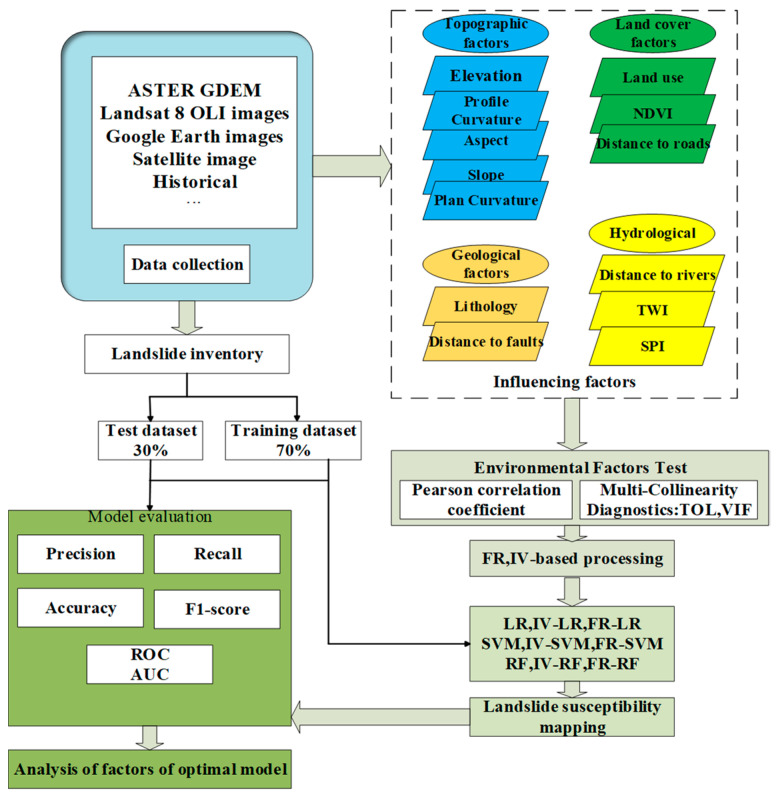
Methodological flowchart.

**Figure 5 sensors-23-02549-f005:**
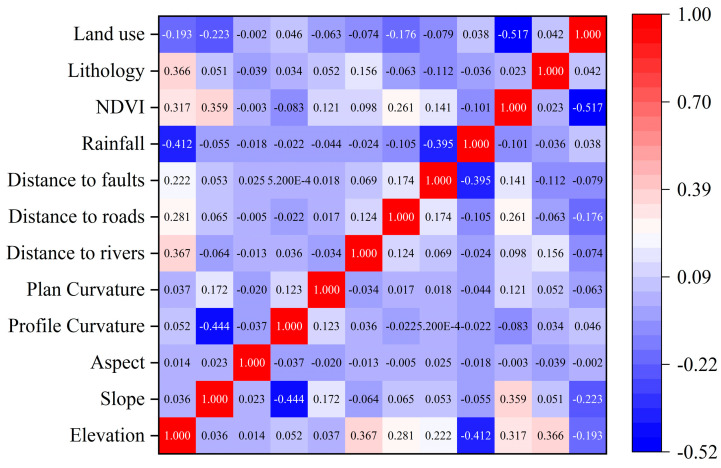
Pearson correlation values between factors.

**Figure 6 sensors-23-02549-f006:**
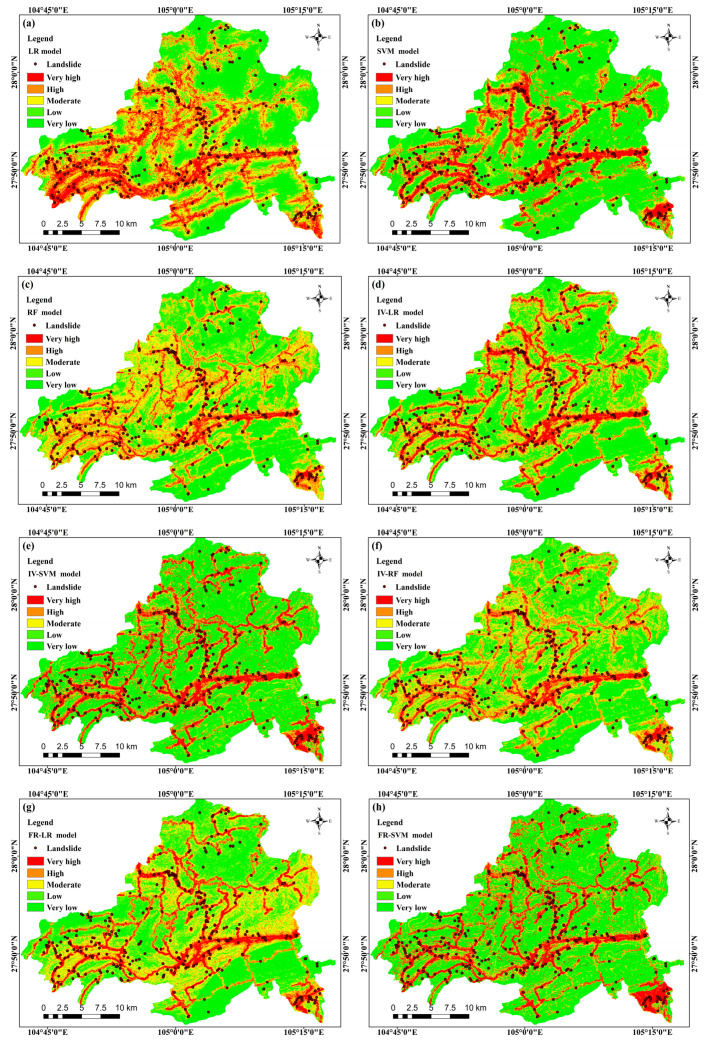
Landslide susceptibility mapping of different models: (**a**) LR, (**b**) SVM, (**c**) RF, (**d**) IV–LR, (**e**) IV–SVM, (**f**) IV–RF, (**g**) FR–LR, (**h**) RF–SVM, and (**i**) FR–RF.

**Figure 7 sensors-23-02549-f007:**
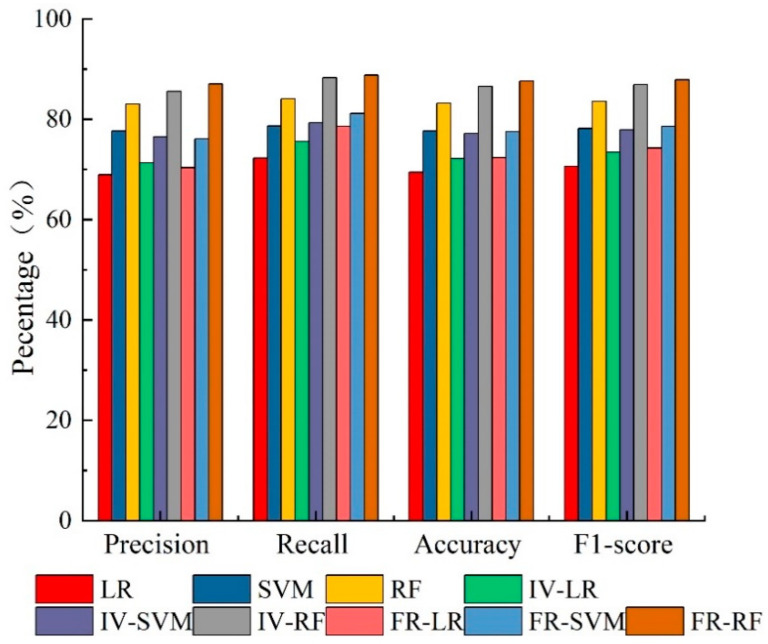
Precision comparsion of the model.

**Figure 8 sensors-23-02549-f008:**
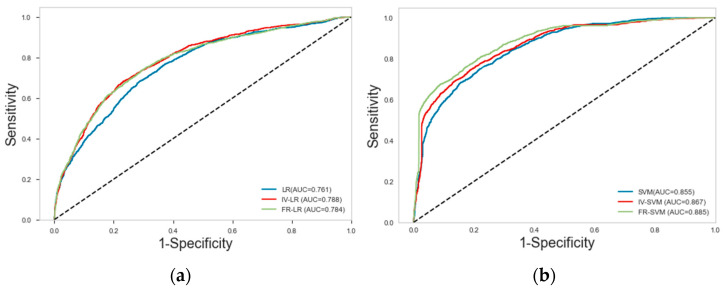
ROC curves with associated AUC value validation set: (**a**) LR, IV–LR, and FR–LR; (**b**) SVM, IV–SVM, and FR–SVM; (**c**) RF, IV–RF, and FR–RF; and (**d**) LR, SVM, and RF.

**Figure 9 sensors-23-02549-f009:**
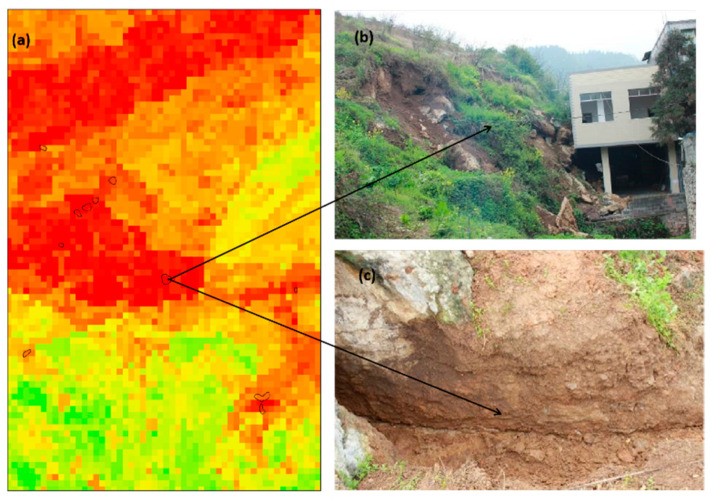
Landslide susceptibility prediction and case validation and analysis: (**a**) Simple labeling based on the FR–RF model. (**b**) General picture of Longdongyan landslide. (**c**) Details of landslides.

**Figure 10 sensors-23-02549-f010:**
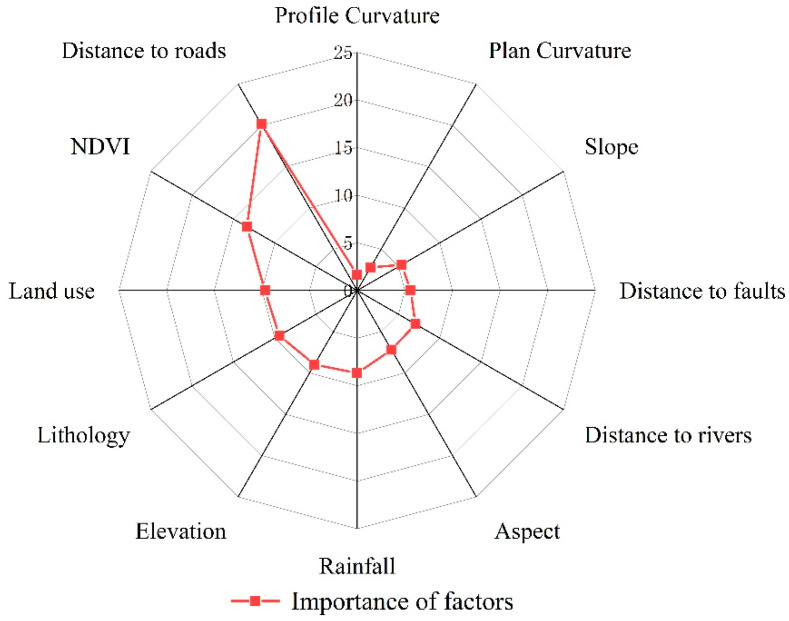
Importance ranking of environmental factors of the FR–RF model.

**Table 1 sensors-23-02549-t001:** Landslide genesis factors and their sources.

Factors	Clusters	Sources
Elevation	Topographic	ASTER GDEM (spatial resolution of 30 m × 30 m) (http://www.gscloud.cn/, accessed on 13 May 2021)
Slope		
Aspect		
Plan curvature		
Profile curvature		
Distance to faults	Geological	Geological map of China (Scale of 1:20,000)
Lithology		
Rainfall	Hydrological	Data Center of the Chinese Academy of Sciences (Spatial resolution 1 km × 1 km) (http://www.resdc.cn, accessed on 15 July 2021)
Distance to rivers		The thematic map of the river system in China from 91 satellite map assistant software (Scale of 1:50,000)
NDVI	Land cover	The geospatial data cloud network (The Landsat 8 OLI image on http://www.gscloud.cn/ (accessed on 2 August 2021))
Land use		The land use and land cover change database in China (http://www.resdc.cn, accessed on 10 September 2021)
Distance to roads		Open street map data (https://www.openstreetmap.org, accessed on 3 August 2021)

**Table 2 sensors-23-02549-t002:** Confusion matrix.

Prediction Situation	Actual Situation	
	Positive Sample	Negative Sample
Landslide	True positive (TP)	False positive (FP)
Negative sample	False negative (FN)	True negative (TN)

**Table 3 sensors-23-02549-t003:** Collinearity diagnostic results of influence factors.

Factors	TOL	VIF
Elevation	0.513	1.951
Slope	0.844	1.185
Aspect	0.996	1.004
Profile curvature	0.713	1.403
Plan curvature	0.718	1.392
Distance to rivers	0.838	1.193
Distance to roads	0.853	1.172
Distance to faults	0.800	1.25
Rainfall	0.702	1.424
NDVI	0.603	1.659
Lithology	0.779	1.283
Land use	0.721	1.388

**Table 4 sensors-23-02549-t004:** Classification of attribute intervals of environmental factors with information values and frequency ratios.

Factors	Classes	FR	IV
Elevation/m	478–800	1.66	0.51
	801–900	0.68	−0.38
	901–1000	0.97	−0.03
	1001–1100	1.07	0.06
	1101–1200	0.97	−0.03
	1201–1300	1.53	0.43
	1300–1500	0.9	−0.1
	>1500	0.33	−1.12
Aspect	Flat (−1)	0	−9.94
	North (0–22.5, 337.5–360)	0.88	−0.25
	Northeast (22.5–67.5)	0.93	−0.08
	East (67.5–11.25)	1.05	0.05
	Southeast (112.5–157.5)	0.98	−0.02
	South (157.5–202.5)	1.25	0.23
	Southwest (202.5–247.5)	1.2	0.18
	West (247.5–292.5)	1.05	0.05
	Northwest (292.5–337.5)	0.83	−0.19
NDVI	−0.71	15.03	2.71
	0.16–0.34	4.59	1.52
	0.34–0.47	1.63	0.49
	0.47–0.77	1.36	0.31
	0.77–0.92	0.5	−0.69
Distance to faults/m	0–200	1.69	0.52
	200–400	1.97	0.68
	400–600	1.69	0.53
	600–800	1.16	0.15
	800–1000	0.94	−0.06
	1000–1200	1.22	0.2
	>1200	0.87	−0.14
Profile curvature	−17.95–−8.8	0	−9.73
	−8.8–0.35	0.88	−0.13
	0.35–4.92	1.24	0.21
	4.92–9.50	1.55	0.44
	9.50–23.23	0	−10.79
Plan curvature	−12.33–−0.15	1.08	0.08
	−0.15–0.29	0.94	−0.07
	0.29–1.49	0.96	−0.05
	1.49–2.47	0.94	−0.06
	2.47–15.42	1.06	0.06
Slope	0–10	0.91	−0.1
	10–20	0.97	−0.03
	20–25	1.02	0.02
	25–30	1.07	0.07
	30–40	1.09	0.08
	40–45	0.91	−0.09
	45–76	0.8	−0.23
Rainfall/mm	1076.24–1080.39	0.86	−0.16
	1080.39–1082.46	0.42	−0.91
	1082.46–1084.53	0.2	−1.64
	1084.53–1088.97	1.43	0.42
	1088.97–1095.68	1.15	0.2
	1095.68–1099.62	1	−0.01
	1099.62–1101.40	0.88	−0.15
Distance to roads/m	0–200	3.02	1.1
	200–400	1	0
	400–600	0.73	−0.32
	600–800	0.42	−0.86
	800–1000	0.56	−0.57
	>1000	0.35	−1.05
Distance to rivers/m	0–200	1.44	0.36
	200–400	1.01	0.01
	400–600	0.71	−0.35
	600–800	1.12	0.11
	800–1000	0.56	−0.58
	>1000	0.91	−0.09
Land use	Forestland	0.78	−0.25
	Farmland	0.83	−0.19
	Residential areas	4.65	1.54
	Grassland	1.68	0.52
	Water	2.62	0.96
	Bareland	9.73	2.28
	Gardenland	1.3	0.26
Lithology	Dolomite	1.74	0.56
	Mudstone and limestone	0.44	−0.82
	Shales	1.4	0.34
	Magmatic veins	1.25	0.22
	Metamorphic rock	2.45	0.9
	Granitic rocks	0.74	−0.3

**Table 5 sensors-23-02549-t005:** Coefficients and constant terms for LR, IV–LR, and FR–LR.

Figure	LR	IV–LR	FR–LR
Elevation	0	0.28	0.36
Slope	0.02	2.21	2.1
Aspect	0	1.12	1.08
Distance to rivers	0	−0.374	−0.421
Distance to roads	−0.001	0.761	0.598
Distance to faults	0	0.298	0.288
Constant	15.1	−0.1	−6.5
Profile curvature	1.072	1.802	1.182
Plan curvature	1.108	0.575	0.555
Rainfall	−0.013	0.642	0.791
NDVI	−3.38	0.667	0.237
Lithology	0.158	0.631	0.656
Land use	0.065	0.343	0.182

**Table 6 sensors-23-02549-t006:** Distribution of landslides at all susceptibility levels with different models.

Model	Geohazard Level	Number of Area Pixels	Area Pixels of Percentage (%)	Number of Landslide Pixels	Ratio of Landslides (%)	Frequency (FR)
LR	Very low	274,712	17.74	290	3.96	0.22
	Low	426,520	27.55	578	7.9	0.29
	Moderate	416,408	26.9	1424	19.46	0.72
	High	281,493	18.18	2203	30.1	1.66
	Very high	149,072	9.63	2824	38.58	4.01
IV–LR	Very low	393,109	25.39	278	3.8	0.15
	Low	479,922	31	550	7.51	0.24
	Moderate	339,148	21.91	1262	17.24	0.79
	High	212,188	13.71	2235	30.54	2.23
	Very high	133,835	8.64	2994	40.91	4.73
FR–LR	Very low	397,363	25.67	280	3.83	0.15
	Low	479,757	30.99	560	7.65	0.25
	Moderate	344,913	22.28	1266	17.3	0.78
	High	192,377	12.43	2275	31.08	2.5
	Very high	133,794	8.64	2938	40.14	4.65
SVM	Very low	572,224	36.96	247	3.37	0.09
	Low	373,936	24.15	702	9.59	0.4
	Moderate	218,640	14.12	908	12.41	0.88
	High	244,536	15.79	1318	18.01	1.14
	Very high	138,868	8.97	4144	56.62	6.31
IV–SVM	Very low	677,668	43.77	263	8.84	0.2
	Low	382,465	24.7	824	9.89	0.4
	Moderate	172,218	11.12	684	9.35	0.84
	High	180,610	11.67	1372	14.31	1.23
	Very high	135,244	8.74	4176	57.62	6.6
FR–SVM	Very low	652,793	42.16	210	2.87	0.07
	Low	374,546	24.19	820	11.2	0.46
	Moderate	173,883	11.23	686	9.37	0.83
	High	180,751	11.67	1423	19.44	1.67
	Very high	136,231	8.8	4180	57.11	6.49
RF	Very low	272,850	17.62	48	0.66	0.01
	Low	429,932	27.77	346	4.73	0.07
	Moderate	413,439	26.7	978	13.36	0.4
	High	306,267	19.78	1727	23.6	1.51
	Very high	125,717	8.12	4220	57.66	7.05
IV–RF	Very low	368,515	23.8	37	0.51	0.02
	Low	470,434	30.39	372	5.08	0.17
	Moderate	342,154	22.1	924	12.62	0.57
	High	238,676	15.42	1782	24.35	1.58
	Very high	128,419	8.29	4202	57.41	6.92
FR–RF	Very low	385,216	24.88	36	0.49	0.02
	Low	471,985	30.49	325	4.44	0.15
	Moderate	333,961	21.57	874	11.94	0.55
	High	230,193	14.87	1758	24.02	1.62
	Very high	126,850	8.19	4326	59.11	7.21

**Table 7 sensors-23-02549-t007:** Analysis of prediction ability of different models by validation samples.

	LR	SVM	RF	IV–LR	IV–SVM	IV–RF	FR–LR	FR–SVM	FR–RF
TP	1611	1753	1874	1685	1768	1968	1752	1809	1979
TN	1438	1658	1779	1485	1621	1831	1424	1596	1868
FP	725	505	384	678	542	332	739	567	295
FN	618	476	355	544	461	261	477	420	250
Precision (%)	68.96	77.64	82.99	71.31	76.54	85.57	70.33	76.14	87.03
Recall (%)	72.27	78.65	84.07	75.59	79.32	88.29	78.60	81.16	88.78
Accuracy (%)	69.42	77.66	83.17	72.18	77.16	86.50	72.31	77.53	87.59
F1 score (%)	70.58	78.14	83.53	73.39	77.90	86.91	74.24	78.57	87.90

## Data Availability

The data that support the findings of this study are available on request from the authors.

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
