# Peer review of "Landslide Susceptibility Evaluation of Machine Learning Based on Information Volume and Frequency Ratio: A Case Study of Weixin County, China"

_sensors, 2023, doi:10.3390/s23052549_

Round 1
Reviewer 1 Report
- This study is a case study of landslide susceptibility evaluation with statistical models coupling machine learning models. It is a trend of researches on landslide prediction in recent.
- However, the reviewer thinks that this study analyzes landslide susceptibility based on only statistical methods without geologically based concepts and meanings. It looks like a routine work of data processing with multiple statistical and machine learning models. For example, this study builds nine prediction models based on statistical models coupling with machine learning models. Instead of the building the prediction models routinely, it is recommended that the analysis is to perform based on some appropriate methods which consider the intrinsic characteristics of landslide influential factors both in statistical and geological concepts. It would be more meaningful research paper if the authors explain a selection process of the appropriate methods in the study.
- The authors explain the analysis results in terms of only statistical values. However, the results need to be explained and discussed in geological meanings, especially targeted to characteristics of landslide occurrence.
- In table 2, the authors classified the lithological information based on the geological age of the study area. The reviewer wonders whether the geological age is influential to landslide triggering or not. The geological age is a concept of chronology to classify geological strata. However, landslides are usually influenced by physical and chemical factors as well as hydrological factors, not by chronological factor. It is recommended to reconsider the classification of lithology information based on the geological age. If possible, lithology information is necessary to be classified dependent on rock sorts.
- It is not necessary to explain basic theory of a method in detail. Therefore, chapters from 3.2 to 3.5 are recommended to reduce the contents. The authors should write related references when explain the basic theory. In the current manuscript, references are not written enough in the chapters from 3.2 to 3.5.
- The authors use "Disaster" mixed with "Hazard" in the manuscript. Because they have different fundamental meaning, the authors should use the terminology carefully based on exact meaning.
- Some sentences are not written perfectly in English grammar. Some numbers of equations seem to be written wrong. Please check them out and correct the errors.
Author Response
Dear reviewer:
Thanks very much for taking your time to review this manuscript. I really appreciate all your comments and suggestions!
Question 1: It is recommended that the analysis is to perform based on some appropriate methods which consider the intrinsic characteristics of landslide influential factors both in statistical and geological concepts. It would be more meaningful research paper if the authors explain a selection process of the appropriate methods in the study.
Thanks for the reviewer's advice, but I still have some inthorough understanding of the combination of geological concepts and statistics, and lack of sufficient understanding of geological knowledge. My current understanding is that the evaluation and prediction of landslide susceptibility in a large range of areas are commonly used. The superposition analysis based on statistical methods is commonly used at present, but the statistical methods have certain defects, and the deficiencies of statistical methods avoided by machine learning methods to a certain extent make landslide prediction reasonable. Although various methods and models are emerging in an infinite number, However, due to the different complexity of different areas, a landslide evaluation and prediction method suitable for all areas has not been found. In the next step, we will follow the suggestions of reviewers, synthesize statistics and consider geological concepts to enrich the current deficiencies in this direction. Thanks again for the reviewer's advice.
Question 2: The authors explain the analysis results in terms of only statistical values. However, the results need to be explained and discussed in geological meanings, especially targeted to characteristics of landslide occurrence.
At present, the research in this paper is only limited to statistical explanation in terms of the interpretation of results. Due to the limitations of the data obtained at present, the lack of time for field research, and the imperfect data of landslide types in the study area, there is no landslide induced by rainfall or caused by earthquake. In various landslide types, the time for field investigation is insufficient. As a result, the current study is unable to provide a more reasonable explanation for the characteristics of landslide occurrence. In future studies, we will devote ourselves to explaining and discussing the rationality and accuracy of the results of this study from the geological sense. Again, I apologize for my lack of knowledge in the research.
Question 3: If possible, lithology information is necessary to be classified dependent on rock sorts.
According to suggestions of reviewers, we classified the stratigraphic lithology from the perspective of lithology, and classified the stratigraphic lithology in the study area into: Dolomite, Mudstone and limestone, Shales, Magmatic veins, Metamorphic rock, Granitic rocks.
Question 4:It is not necessary to explain basic theory of a method in detail. Therefore, chapters from 3.2 to 3.5 are recommended to reduce the contents. The authors should write related references when explain the basic theory. In the current manuscript, references are not written enough in the chapters from 3.2 to 3.5.
In sections 3.2 to 3.5, the method lacks references, we added some references, but for the deletion of this section, we comprehensively considered the deletion of this part, but maybe the deletion did not meet the reviewer's requirements. If the deletion did not meet the reviewer's requirements, we feel sorry for that, and we will continue to improve this part of content.
Question 5: The authors use "Disaster" mixed with "Hazard" in the manuscript. Because they have different fundamental meaning, the authors should use the terminology carefully based on exact meaning.
In view of the meaning of the word disaster and hazard in the article, we changed the definition to danger after referring to relevant literature.
Question 6: Some sentences are not written perfectly in English grammar. Some numbers of equations seem to be written wrong. Please check them out and correct the errors.
The grammar of the needle part is not perfect, so we have modified it. We have corrected the wrong part of the equation and checked the rationality of all formulas in this paper.
Sincerely,
Wancai He
Author Response
Dear reviewer:
Thanks very much for taking your time to review this manuscript. I really appreciate all your comments and suggestions! Please find my itemized responses in below and my revisions in the re-submitted files.
Question 1 : Abstract:The content is too comprehensive and it appears too much content, innovation points could be further refined.
In view of the excessive content of the abstract and the innovation points are not prominent, the abstract has been reorganized and modified, and the content and innovation points of the abstract have been refined.
Question 2 : There is less verifiable data, which can increase the test data, and improve the scientificity of Landslide susceptibility evaluation of machine learning. Overall, it was very well done.
I'm sorry that currently, due to the conditions and time of data acquisition, there are few data for verification. In the future research, we will add more verification data to make the research direction more meaningful. In the next step, our research will obtain higher resolution and more detailed data, strengthen the verification of the evaluation results, and make the evaluation results more accurate and have more practical reference value. The verification results are more reasonable.
Question 3: The discussion is too simple, and the result description focuses on details, resulting in insufficient summary of innovation points. A better summary would help readers understand the innovation of this study.
In view of the lack of discussion content, in the fifth part, the discussion content is added and marked with red font. There are too many details in the conclusion part. We have summarized the whole paper and enriched the conclusion content, highlighting the innovation of this paper. The modification of the conclusion part has been marked in red font.
Sincerely,
Wancai He
Round 2
Reviewer 1 Report
The reviewer has commented that the authors need to analyze the results based on the geological concepts and meanings, not by only statistical approaches. However, the reviewer cannot find an appropriate revision or efforts to apply the reviewer's comments in the revised manuscript. Because the target of the analysis is landslide, the statistical results should be analyzed and discussed by geological knowlegment, not only by a statistical methods. Therefore, the reviewer strongly recommends the authors to revise the manuscript based on the reviewer's commnets.
Author Response
Dear reviewer:
Thanks again for the reviewer's valuable comments on the manuscript. I am sorry that my first revision did not meet the reviewer's requirements.I modified the questions raised by the reviewer again.
Question:The authors need to analyze the results based on the geological concepts and meanings, not by only statistical approaches. .
The revised part is marked in red in the revised manuscript. In view of the modification of this problem, I analyzed the stratigraphic lithology of landslide in extremely high and highly prone areas from the perspective of lithology. This paper analyzes the occurrence of landslides in the study area under geological conditions and enriches the results of landslide susceptibility mapping. However, due to my lack of geological knowledge, the modification may not be complete. I'm sorry again.Due to my lack of geological knowledge and incomplete collection of geological data in the study area, I failed to conduct in-depth geological analysis of the landslide susceptibility results as required by the reviewer, but only analyzed them from the perspective of statistics. Aiming at the current hot spot of landslide susceptibility research, this paper selected the currently commonly used machine learning model and statistical model for coupling research, and verified the reliability of the model to a certain extent. In the study, this paper laid emphasis on the study of methods, which led to some problems in the interpretation of the results. Here, I would like to express my special thanks to the reviewer for raising this question. It made me realize that the study of landslide susceptibility evaluation should not only focus on the study of current model methods, but also focus on the study of the mechanism and direction of landslide occurrence. Only by understanding these aspects, can we further predict the occurrence of landslide and make our research more meaningful. The use of geological knowledge to analyze and discuss landslide proposed by the reviewer also provides a new idea and target for me to continue the research of landslide susceptibility evaluation in the future. My previous research has always focused on the study of landslide susceptibility evaluation model, ignoring that the object of our study is landslide. When we analyze results and predict landslide, we should not only focus on the study of the model. It is also necessary to study the mechanism and nature of landslide occurrence, so as to make the prediction of regional landslide more reasonable and reliable. At present, the main method for predicting regional landslide is to make landslide susceptibility mapping by using artificial intelligence method, and to prevent regional landslide according to the classification of landslide susceptibility. This method is more popular in the current research. Although a lot of research has been conducted, the current research has not shown that a certain model can be applied to all types of landslide, so this paper focuses on the research of methods, studying the artificial intelligence method in landslide prediction in different areas. In terms of the discussion of the results, analysis and verification are carried out according to the geological knowledge. The geological environment selected in this paper, such as the formation lithology and the distance from the fault zone, is only an environmental factor, without detailed classification of landslide rainfall landslide and earthquake landslide. It is only based on the location of the existing landslide and the occurrence of environmental factors to predict the possibility of landslide in the future. However, according to the reviewer's opinion, we will collect more detailed geological data and detailed classified landslide type data in the study area in the future research, and apply the method of artificial intelligence and geological knowledge to the landslide susceptibility evaluation. The next research will pay more attention to the study of geological knowledge in the evaluation results. In this revision, geological knowledge has been analyzed in terms of evaluation results by referring to relevant geological data collected, but there are still many shortcomings that need to be improved. We hope that the reviewers can give us valuable comments again. We will continue to revise the manuscript according to the reviewers' comments. In the next study, we will continue to improve our shortcomings in landslide susceptibility evaluation according to the suggestions of reviewers.
Sincerely,
Wancai He
Round 3
Reviewer 1 Report
The reviewer recommends the authors to perform further studies considering analysis of the research results with geological concepts in the future.
Author Response
Dear Reviewers.
Thank you very much for your comments on the manuscript. Based on your suggestions, we have tried our best to revise the relevant sections and have made some changes to the manuscript. These changes do not affect the content or the framework of the paper. All your questions are answered below. Here we have listed the changes and marked them in red in the revised paper.
According to your question: consider further research findings and analysis of geological concepts in the future. Results (2) and (3) analyse landslide predictions in the study area from the perspective of stratigraphic lithology and analyse in detail the lithology of very high susceptibility areas from the perspective of stratigraphic lithology. However, the geological information collected in this paper is not very detailed and there is not enough time for fieldwork, so there are some shortcomings in the analysis of the results. Secondly, in the accuracy analysis section, we added landslide interpretation and survey data from 2020-2021, selected the RF-RF model to analyse and verify the reliability of the model used, and selected typical landslides from this period for case studies to analyse the occurrence of landslides in terms of more detailed stratigraphic lithology. The predictive accuracy of the model was also verified. These variations may still be inadequate. Therefore, in the discussion section (1), based on the reviewers' suggestions, we propose that in future studies, we consider analysing the results of landslide susceptibility mapping from the perspective of geological concepts to make the results more accurate and reliable. In this paper, we offer an outlook on future research on landslide susceptibility and propose that in future research
In future research, we will follow the reviewer's recommendations and requirements to analyse landslide susceptibility mapping studies from a geological concept. We will strive to achieve breakthroughs and innovative results in future research. Once again, we thank the reviewers for their comments and valuable advice, which also provide us with breakthrough directions and research priorities for future research.
We sincerely thank the reviewers for their enthusiastic work. With all the comments and suggestions, the quality of my paper has improved significantly. We hope that the revision will be approved. If you have any questions, please do not hesitate to contact us. Thank you again for your comments and suggestions.
Sincerely
He Wancai